# A multiplexed, single-cell sequencing screen identifies compounds that increase neurogenic reprogramming of murine Muller glia

Amy Tresenrider[1], Marcus Hooper[2], Levi Todd[2], Faith Kierney[2], Nicolai A Blasdel[2], Cole Trapnell[1,3,4]*, Thomas A Reh[2]*

[1]Department of Genome Sciences, University of Washington, Seattle, United States; [2]Department of Biological Structure, University of Washington, Seattle, United States; [3]Brotman-Baty Institute for Precision Medicine, University of Washington, Seattle, United States; [4]Allen Discovery Center for Cell Lineage Tracing, Seattle, United States

## eLife assessment

This manuscript used the sci-Plex system for screening compounds to improve the Ascl1-induced reprogramming from Müller glia to bipolar neurons in vitro, followed by in vivo characterization of two promising compounds in mice. The findings are **valuable** for future studies to develop cell replacement strategies for treatment of retinal degeneration. The strength of evidence is **solid**, featuring a scalable drug screening design, albeit with limited mechanistic insights.

*For correspondence:
coletrap@uw.edu (CT);
tomreh@u.washington.edu (TAR)

**Abstract** Retinal degeneration in mammals causes permanent loss of vision, due to an inability to regenerate naturally. Some non-mammalian vertebrates show robust regeneration, via Muller glia (MG). We have recently made significant progress in stimulating adult mouse MG to regenerate functional neurons by transgenic expression of the proneural transcription factor Ascl1. While these results showed that MG can serve as an endogenous source of neuronal replacement, the efficacy of this process is limited. With the goal of improving this in mammals, we designed a small molecule screen using sci-Plex, a method to multiplex up to thousands of single-nucleus RNA-seq conditions into a single experiment. We used this technology to screen a library of 92 compounds, identified, and validated two that promote neurogenesis in vivo. Our results demonstrate that high-throughput single-cell molecular profiling can substantially improve the discovery process for molecules and pathways that can stimulate neural regeneration and further demonstrate the potential for this approach to restore vision in patients with retinal disease.

## Introduction

In mammalian species, retinal neurodegeneration results in permanent visual impairment because the central nervous system (CNS) lacks an endogenous regenerative capacity. In contrast, many non-mammalian vertebrates can replace lost neurons via neurogenic reprogramming of non-neuronal cells (*Todd and Reh, 2022*). Efforts to stimulate neural cell replacement in mammals, akin to what occurs in regenerative species, have been met with some success (*Jorstad et al., 2017*; *Lentini et al., 2021*). These approaches are typically focused on using transcription factors (TFs) to reprogram glial cells into

neurogenic precursors in contexts across the CNS (brain, retina, and spinal cord). However, the reprogramming is incomplete, affecting only a subset of cells or producing cells without mature neuronal markers. Given the vast array of small molecules and biologics available, there is potential to screen for factors that improve such gene therapy techniques. Scalable methods capable of screening an increased number of reprogramming treatments have the potential to uncover factors that improve the regenerative potential of the mammalian CNS. These efforts may lead to future cell-replacement therapeutics.

We have recently developed methods to stimulate the regeneration of neurons from Muller glia (MG) in the mammalian retina in vitro and in vivo (*Jorstad et al., 2017*; *Pollak et al., 2013*; *Jorstad et al., 2020*; *Todd et al., 2021*; *Todd et al., 2022*; *Ueki et al., 2015*). The in vitro viral overexpression of Ascl1 in primary MG is able to reprogram glia from young mice to acquire a neurogenic progenitor state and generate neurons (*Pollak et al., 2013*; *Gascón et al., 2016*), and similar results have been obtained with brain astrocytes (*Pollak et al., 2013*; *Gascón et al., 2016*). Ascl1 can also stimulate a neurogenic state in MG in vivo using a transgenic mouse line that targets this TF specifically to MG. When these mice are subjected to retinal injury using a neurotoxic dose of NMDA (N-methyl-D-aspartate) and subsequently treated with histone deacetylase (HDAC) inhibitors, the MG regenerate functional neurons in adult mice (*Jorstad et al., 2017*). Follow-up studies combining Ascl1 with other TFs have proven useful in stimulating glia to regenerate alternate arrays of neuronal fates (*Todd et al., 2021*; *Todd et al., 2022*). These findings have established glia as a promising source for new neurons that could potentially be used to treat neurodegenerative disorders; however, the efficacy of neurogenesis is often too low to provide enough cells for functional recovery. Several studies have shown that cell signaling molecules can also affect the process of reprogramming cell fates, and investigators have suggested that small molecules might be useful in optimizing this approach for clinical applications of neural repair in combination or separate from TF overexpression (*Jorstad et al., 2020*; *Gascón et al., 2016*; *Janowska et al., 2019*; *Cheng et al., 2015*; *Gao et al., 2017*).

To identify additional compounds that aid in the reprogramming process and further stimulate neurogenesis from MG-derived progenitors, we set forth to design an assay that could scale the number of small molecules tested in a given in vitro experiment. Several limitations had made such plate-based screening difficult in the past: (1) screening was performed using marker genes and required the design of specific mice or cell lines to detect the emergence of only a single-cell type; (2) MG are not a large population of cells in the retina of young mice, thus the starting material is limited; and (3) current imaging-based methods to quantify the reprogramming outcome are low throughput. While traditional single-cell sequencing methods enable the molecular characterization of all cell types in a culture using low cell input without the need for marker genes, they are too costly for most labs to perform for more than a handful of samples. sci-Plex, a combinatorial indexing-based technology, scales the number of samples within a given single-cell sequencing experiment to up to thousands of samples, overcoming such limitations (*Srivatsan et al., 2020*; *Srivatsan et al., 2020*; *Cao et al., 2017*). However, current applications of sci-Plex have been performed in combination with the high-throughput sci-RNA-seq3 protocol using culturing systems or organisms in which cell number is not limited (*Srivatsan et al., 2020*; *Saunders et al., 2022*; *Dorrity et al., 2022*; *Tresenrider et al., 2023*). Here, we demonstrate that sci-Plex is also compatible with lower input samples in combination with the sci-RNA-seq2 protocol which requires a lower cell input and has a higher cell recovery rate. Our application of sci-Plex to reprogramming MG is the first demonstration of sci-Plex's utility for screening chemical compounds in a low abundance primary cell type with disease relevance. To examine and validate the utility of sci-Plex for this purpose, we applied sci-Plex to MG exposed to Ascl1 overexpression for different timings and durations. We saw that, increasing the length of Ascl1 exposure increased the number of neurons produced, and we uncovered molecular features associated with progression toward the neuronal fate. When we applied sci-Plex to a small molecule screen for factors associated with stem cell biology, we uncovered compounds that had effects on neurogenesis and total cell recovery. Several of the most effective compounds identified in vitro were further tested in vivo in a reprogramming assay where Ascl1 is over-expressed specifically in MG (i.e. ANT protocol). The results of this secondary in vivo screen identified two compounds that significantly enhanced neurogenesis in adult mouse retina.

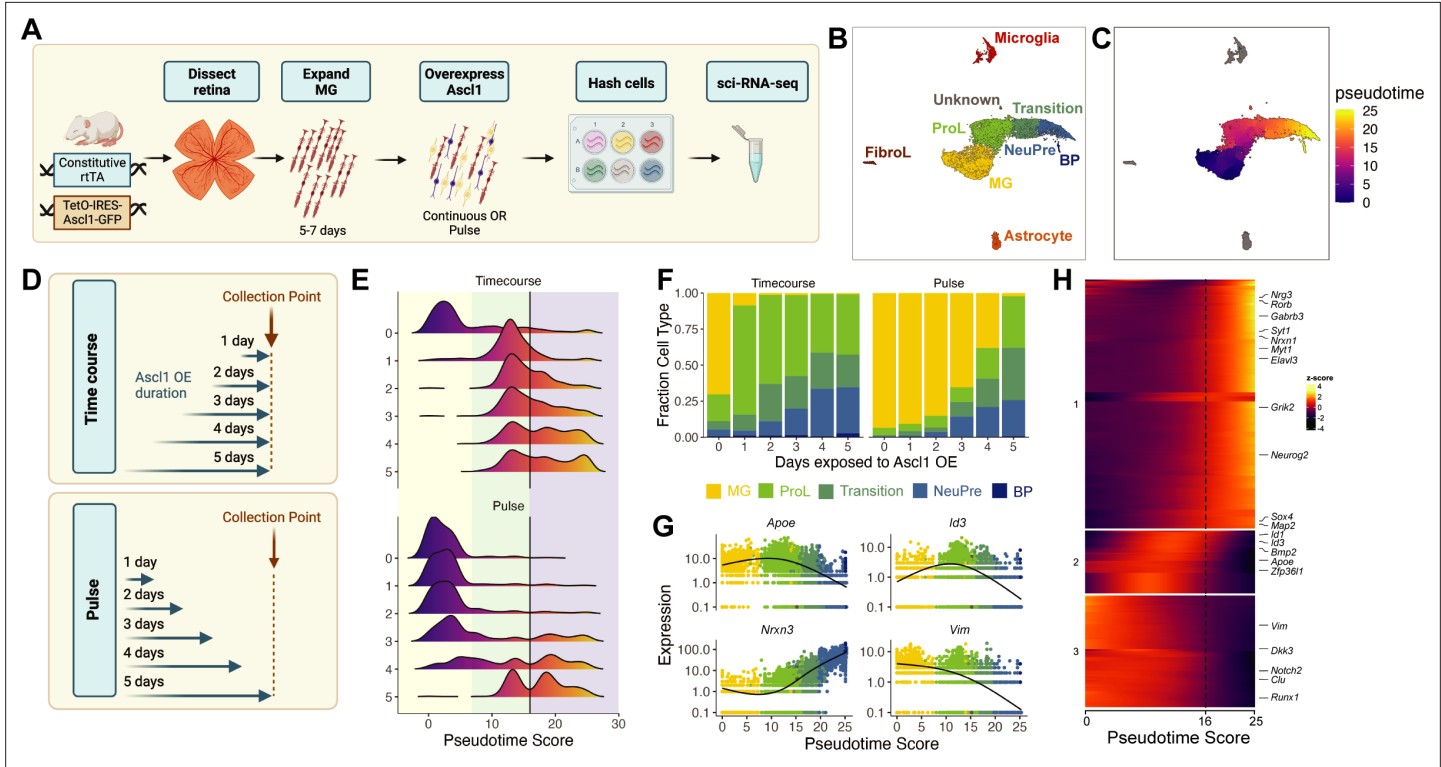

**Figure 1.** sci-Plex captures the in vitro temporal dynamics of neurogenic reprogramming from Muller glia (MG). (**A**) Schematic of the sci-Plex experimental design for assaying reprogramming MG isolated from P11 mice. (**B**) Combined UMAP of cells from the Timecourse and Pulse experiments. Cells are colored by cell type. (**C**) UMAP displaying the pseudotime scores that were calculated for the MG to Bipolar trajectory. (**D**) Schematic depicting the timing and duration of Ascl1 OE in the Timecourse (*n* = 2 wells per condition) and Pulse (*n* = 3 wells per condition) experiments. (**E**) Histograms displaying the frequency of cells with each pseudotime score across Ascl1 OE conditions. The vertical black lines are at pseudotime score 16. The yellow region corresponds to the MG, the green region corresponds to the ProL cells, and the purple region reflects cells from the Transition to BP cell state. (**F**) Stacked bar plot of the cell type composition across all Ascl1 OE conditions. The colors represent the cell types as indicated. Only the MG and MG-derived cell types are included. (**G**) Gene expression plots along pseudotime for genes of interest. Each point represents an individual cell's expression of the indicated gene. The cells are colored by cell type as in F. (**H**) Gene expression heatmap for the top 250 DEGs as assessed across a pseudotime score of 10–20. The dashed line is at pseudotime score 16. All cells with a non-infinite pseudotime score are ordered by pseudotime score along the *x*-axis. Genes were clustered by *k*-means into three clusters. Figure 1A was created with BioRender.com and Figure 1D was also created with BioRender.com.

The online version of this article includes the following source data and figure supplement(s) for figure 1:

**Source data 1.** Genes differentially expressed across pseudotime.

**Figure supplement 1.** Sequencing quality control for the Timecourse and Pulse experiments.

**Figure supplement 2.** Cell type annotation for the Timecourse and Pulse experiments.

**Figure supplement 3.** Visualization of cell composition changes across time in the Timecourse and Pulse experiments.

**Figure supplement 4.** Retention of late pseudotime cells in the Pulse experiment matches loss of *Ascl1* expression.

## Results

### sci-Plex captures the in vitro temporal dynamics of neurogenic reprogramming from MG

We first aimed to assess the ability of sci-Plex to assign cells to specific wells in an expected manner by observing the in vitro kinetics and dynamic gene expression program of Ascl1-dependent MG reprogramming. Retinas were isolated from P11 mice expressing a constitutive rtTA driving a tet-inducible Ascl1-GFP. The retinas were dissociated and grown as previously described to enrich for MG (*Pollak et al., 2013*). We have previously shown that induction of Ascl1, by addition of doxycycline to the medium, induces neurogenesis in the MG, and that the new neurons generated from the reprogrammed MG most closely resemble bipolar neurons (*Pollak et al., 2013*; *Figure 1A*). Taking

advantage of the ease by which condition number can be scaled with sci-Plex, we induced Ascl1 in replicate for 1–5 days at varying windows of time, allowing us to temporally track the conversion of MG to neurons (*Figure 1A, B*). Upon collection, nuclei were isolated in their individual treatment wells, the nuclei were barcoded with well-specific polyadenylated oligonucleotides 'hashes', and then pooled for sci-RNA-seq2 (*Srivatsan et al., 2020*; *Cao et al., 2017*). We recovered 12,850 cells across two experiments and 30 wells (Timecourse: 7004, Pulse: 5846) with a median UMI (unique molecular identifier) count of 3172 and 2398 after all filtering steps, respectively (*Figure 1—figure supplement 1A–K*). The hash recovery rate was 68% for both experiments (*Figure 1—figure supplement 1C, D, G, H*). The experiments were integrated and visualized by UMAP (Uniform Manifold Approximation and Projection) using Monocle3 (*Cao et al., 2017*; *Cao et al., 2019*). The clusters were annotated according to cell type using the expression of known marker genes (*Figure 1—figure supplement 2A–C*; *Cao et al., 2017*; *Cao et al., 2019*). Most cells fell into one of four clusters: (1) MG; (2) Progenitor-like cells (ProL); (3) Transition; and (4) neuronal precursors (NeuPre). In addition, we identified other clusters of non-MG-derived cells, including microglia, astrocytes, and fibroblast-like cells (FibroL), potentially derived from vascular cells (*Figure 1C*).

Having annotated the cells, we next ordered all the cells originating from reprogrammed MG into a pseudotime trajectory to better understand the sequence of gene regulatory events on the path from MG to neuronal identity (*Figure 1D*). We then assessed how the timing and duration of Ascl1 overexpression affected reprogramming progression when Ascl1 was added continuously (Timecourse) or pulsed (Pulse). In the Timecourse experiment, Ascl1 expression was induced for either 0, 1, 2, 3, 4, or 5 days and then the cells were collected (*Figure 1B*). During the timecourse, we saw that Ascl1 stimulated a rapid transition away from MG (i.e. the start of the pseudotime trajectory) that persisted across all time points (*Figure 1E*). As expected, increased duration of Ascl1 overexpression led to cells further along the pseudotime trajectory, consistent with the need for Ascl1 expression to drive cells from the ProL state toward the neuronal state. In the Pulse experiment, we initiated Ascl1 expression in all samples at the same time, but then removed doxycycline after either 1, 2, 3, 4, or 5 days to shutdown Ascl1 overexpression. The cells were all collected 5 days after the start of Ascl1 expression regardless of Ascl1 expression duration (*Figure 1B*, Pulse). When Ascl1 was maintained for the entire 5 day period (*Figure 1D, E*), the distribution of cells across pseudotime was similar to that observed in the Timecourse experiment, with few MG remaining. However, the pseudotime profiles were remarkably different when Ascl1 overexpression was removed for any period of time for all cell types with the exception of NeuPre (*Figure 1—figure supplement 3C*). When Ascl1 was expressed in the cells for 2 or fewer days, almost all of the cells were identified as MG (*Figure 1E, F*). After 3 or more days of Ascl1 expression, transition and neurogenic precursor (NeuPre) cells began to accumulate, though a substantial number of MG were also present even after 4 days of Ascl1 exposure and 1 day without when compared to the Timecourse experiment (*Figure 1F*, *Figure 1—figure supplement 3C*). These results suggest that even after 4 days of Ascl1 expression, removal of Ascl1 allows a subset of the cells to return to a glial state, while another subset of cells that have made it far enough along the reprogramming trajectory are no longer dependent on Ascl1 to remain committed to their neuronal fate.

To further understand the molecular features of the cells as they progress toward a more neuronal fate, we analyzed changes in gene expression around the pseudotime stage associated with a decreased reliance on Ascl1. We found the local minima (10 < pseudotime < 20) of pseudotime density plots from the Pulse experiment to be 16 (*Figure 1—figure supplement 4A*). The accumulation of cells with a pseudotime value of >16 encompasses the population of cells that remain in the NeuPre or Bipolar fate even after removal of Ascl1 induction. Coincidentally, in the Timecourse experiment, decreased *Ascl1* expression is observed at pseudotime values >16 (*Figure 1—figure supplement 4B*). While seemingly surprising, this decrease in *Ascl1* expression with continued exposure to the induction signal is a phenomenon we consistently observe in both our in vitro and in vivo reprogramming paradigms (*Todd et al., 2021*; *Todd and Reh, 2022*; *Todd et al., 2020*). We do not have definitive knowledge of the mechanism behind how Ascl1 expression is repressed, but it is possible that mature neurons have a mechanism to downregulate it.

To find the genes dynamically expressed during the loss of *Ascl1* expression and gain of a more neuronal fate, we performed differential gene expression analysis across pseudotime using cells with pseudotime values from 10 to 20 (*Figure 1—source data 1*). We were particularly interested in identifying transcriptional regulators and signaling molecules. In the top 250 most differentially expressed

genes, three major regulatory patterns (Cluster 1: increasing, Cluster 2: a transient increase, Cluster 3: decreasing) were observed (*Figure 1G, H*). The downregulation of *Id1* and *Id3* in Cluster 2 was consistent with known biology: these factors (1) form heterodimers with bHLH TFs making them incapable of binding their target sites (*Roschger and Cabrele, 2017*) and (2) have been previously implicated in maintaining retinal progenitors and MG in a progenitor state (*Jorstad et al., 2017*; *Pollak et al., 2013*; *Jorstad et al., 2020*; *Todd et al., 2021*; *Todd et al., 2022*; *Ueki et al., 2015*). Genes in Cluster 1, which increase during the progenitor-to-neuron transition, are also associated with the normal development of neurons (*Elavl3*, *Map2*) and with processes such as neuronal maturation and synaptogenesis (*Nrxn3*, *Syt1*). There were also four TFs that increased in their expression right around the loss of Ascl1 dependence: *Sox4*, *Neurog2*, *Myt1*, and *Rorb*. All of these genes are expressed in the retina during the early stages of neurogenesis (*Vasconcelos et al., 2016*; *Lee et al., 2019*; *Hufnagel et al., 2010*; *Liu et al., 2017*; *Usui et al., 2013*; *Jiang et al., 2013*). Of note, *Myt1* is a direct target of Ascl1 and although it has been studied more thoroughly in the brain, it is associated with repression of progenitor gene expression programs (*Vasconcelos et al., 2016*; *Lee et al., 2019*). Another factor on the list, Rorb is involved in the specification of photoreceptor, amacrine, and horizontal but not bipolar cell fates (*Liu et al., 2013*; *Swaroop et al., 2010*). Our results thus show the ability of sci-Plex to be used in MG cultures and we have identified potential regulators of critical steps in the MG to neuron reprogramming paradigm.

## sci-Plex as a screen to identify small molecules that affect Ascl1-dependent MG reprogramming

The above results supported the potential of sci-Plex to study neurogenic reprogramming, so we next embarked on a larger-scale screen for factors that can improve the efficiency of neurogenesis from MG. We screimed to assess the ability of sci-Plex to assign cells to specific wells in an expected manner by observing the in vfrom the TocriScreen Stem Cell Library (*Figure 2—source data 1*). The compounds target a range of biological functions with a bias toward epigenetic modulators and modulators of common signaling pathways (*Figure 2A*). We followed the same treatment regime as for the timecourse experiment in which MG were isolated from P11 mice that express a constitutive rtTA and harbor a tet-inducibe Ascl1-GFP. The MG were selectively cultured before induction of Ascl1 with doxycycline. Small molecules from the TocrisScreen Stem Cell Library were added at the same time as the doxycycline and remained in culture until the cells were collected on Day 5 (*Figure 2B*). The compounds were applied across three doses (0.1, 1, and 10 uM) with each compound being tested on one of two different collection days. We included three wells per collection day in which only doxycycline was added (Only Ascl1) and three wells per collection day in which doxycycline was not added (No Ascl1). To maximize the range of compounds tested, we opted to perform the sci-Plex experiment once, and then follow-up with compounds that elicited strong effects through subsequent in vivo validation experiments. In sum, we screened 92 small molecules and two control conditions from a total of 265 individual wells after filtering out conditions with low cell recovery.

We used sci-RNA-seq2 to profile a total of 27,152 cells (Collection 1: 6282, Collection 2: 20,870) (*Figure 2—figure supplement 1A–G*). The median UMI count was 3551 for Collection 1 and 2166 for Collection 2 (*Figure 2—figure supplement 1H, I*). A hash could be called for 65–74% of the cells (*Figure 2—figure supplement 1C, F*). We performed dimensionality reduction and clustering with Monocle3 and used marker genes to annotate cell types by cluster. All cell types not of retinal/MG origin were discarded (*Figure 2—figure supplement 2A–C*), and the dimensionality reduction and cell type annotation was repeated (*Figure 2C*).

To identify the molecules most likely to promote reprogramming, we pooled the clusters corresponding to neurogenic precursors (NeuPre: *Gap43*, *Snap25*, *Dcx*) and bipolar neurons (BP: *Snap25*, *Vsx2*, *Cabp5*, *Otx2*, *Grik1*) together into a common Neuron annotation and for each drug, calculated the abundance of Neurons relative to Ascl1 (*Figure 2*). Many of the molecules did not have clear effects on the reprogramming process. However, we found several molecules that increased neurogenesis and others that reduced neurogenesis. The molecules that most markedly increased the percentage of Neurons, included DBZ (*Milano et al., 2004*), DMH-1 (*Hao et al., 2010*), RepSox (*Ichida et al., 2009*; *Gellibert et al., 2004*), SB 431542 (*Inman et al., 2002*; *Laping et al., 2002*), and WH-4-023 (*Theunissen et al., 2014*; *Clark et al., 2012*; *Martin et al., 2006*). These compounds target one of three molecular signaling pathways: Notch, BMP/TGF-beta, and LCK/SRC. We also calculated

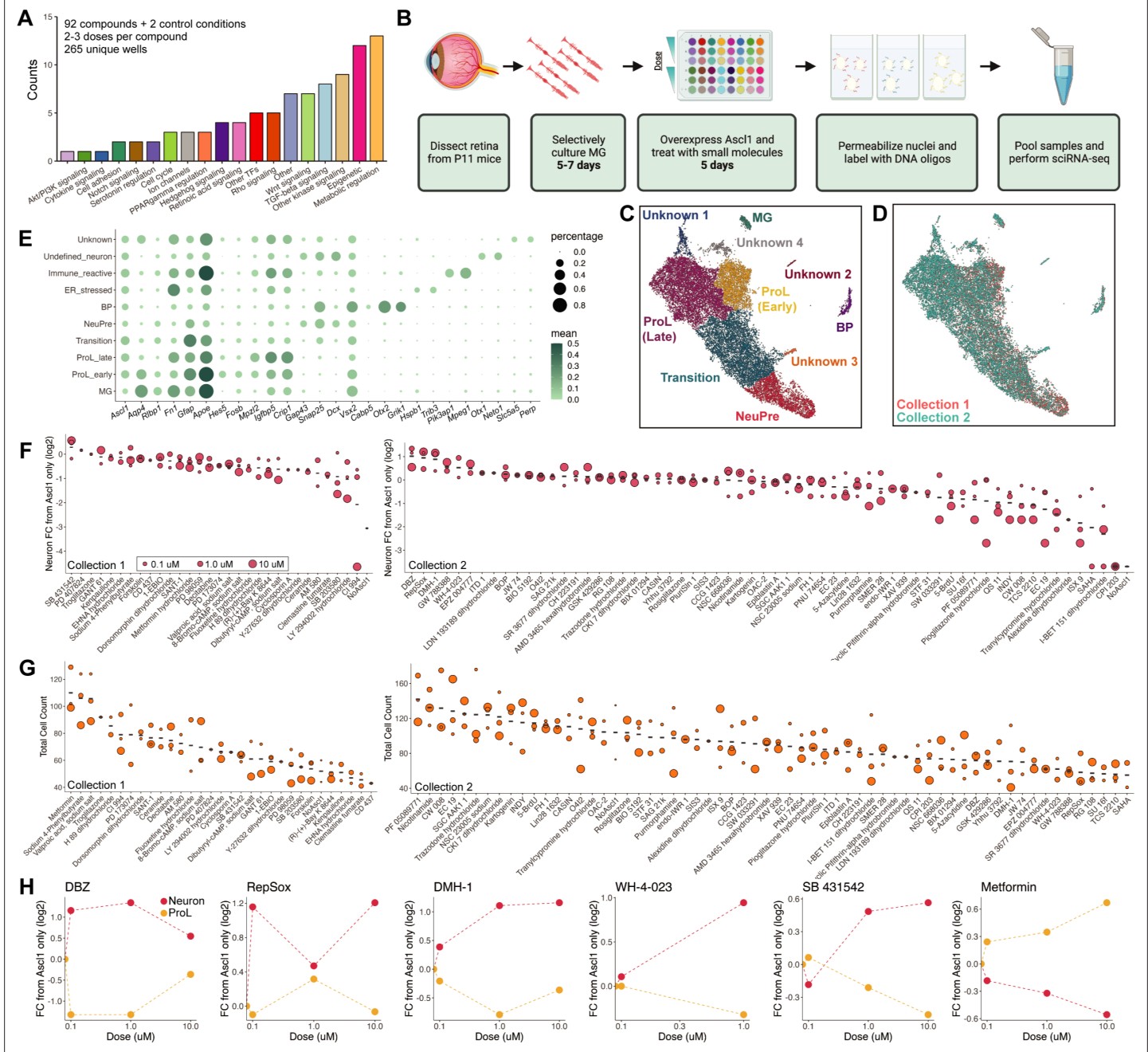

**Figure 2.** sci-Plex as a screen to identify small molecules that affect Ascl1-dependent Muller glia (MG) reprogramming. (**A**) Bar plot representing the distribution of targets for the 92 compounds included in the screen. (**B**) Schematic of the experimental design for the small molecule screen. (**C**) UMAP of the MG and MG-derived cell types. Cells are colored by cell type. (**D**) UMAP from C with cells colored by collection. (**E**) Dot plot of the genes used to define the MG-derived cell types from the screen. Dot size indicates the percent of cells that express the gene of interest. The color indicates the log10 mean UMIs per cell. (**F**) Quantification of the fold change in Neuron cell counts between each indicated condition and the Ascl1 only control. The plots from Collections 1 and 2 used only the control wells collected on their respective days. The dose of each compound is indicated by the size of the dot. (**G**) Quantification of the total cell counts for each treatment. The compound's dose is indicated by the size of the dot. (**H**) Plots displaying the fold change of Neuron (red) and ProL (orange) cell counts compared to Ascl1 only across all doses for the top hits from the screen. All conditions in which at least 20 cells were recovered are displayed. Figure 2B is created with BioRender.com.

The online version of this article includes the following source data and figure supplement(s) for figure 2:

**Source data 1.** Small molecules used from Tocriscreen Stem Cell Library.

**Source data 2.** Genes used and GO annotations in unknown clusters.

*Figure 2 continued on next page*

*Figure 2 continued*

**Figure supplement 1.** Sequencing quality control of small molecule sci-Plex screen.

**Figure supplement 2.** Cell type annotation of small molecule sci-Plex screen.

**Figure supplement 3.** Gene Ontology (GO) term enrichment of Unknown cell clusters from the sci-Plex screen.

**Figure supplement 4.** Cell type distributions across treatment conditions.

the number of cells recovered per well, and noticed that for a few treatments, far more total cells, regardless of cell type identity, were recovered than average. This was most prominent for metformin, sodium 4-phenylbutyrate, and valproic acid (*Figure 2E*). Both sodium 4-phenylbutyrate and valproic acid are HDAC inhibitors which is of interest because TSA (Trichostatin A), another HDAC inhibitor, is a significant stimulus for Ascl1-dependent reprogramming in vivo (*Jorstad et al., 2017*). Unexpectedly, metformin, a compound that is approved for the treatment of diabetes with other potential clinical uses (*Cicero et al., 2012*), also increased cell recovery in our assay. Because metformin has not been evaluated in the context of retinal reprogramming, we decided to pursue it and the five compounds that increased neurogenesis with further experiments in vivo. Thus, the use of sci-Plex aided us in narrowing down 92 compounds to 6 for follow-up investigations.

In addition to detecting shifts along the reprogramming trajectory, we also detected what appeared to be previously undetected cell states (*Figure 2C*, Unknown clusters). To characterize these, we performed Biological Process GO term enrichment on genes with high regional specificity for each cluster (*Figure 2—source data 2*). Cluster Unknown 1 was enriched with genes related to ER stress, Unknown 2 had markers of Immune reactive cells, and Unknown 3 had non-retinal neuronal characteristics (*Figure 2—figure supplement 3A–C*, *Figure 2—source data 2*). The Unknown 4 cells had 77 genes with high specificity to the cluster, but only two enriched gene sets 'regulation of ion transmembrane transport' and 'action potential' (*Figure 2—source data 2*). With such a small number of enriched gene sets, we were unable to make further conclusions about the identity of that cell cluster and have left them annotated as Unknown.

We next set out to understand how each treatment altered the distribution of cells across these novel clusters. We generated a heatmap in which treatments with similar cell type compositions were clustered together (*Figure 2—figure supplement 4A*). In doing this, we found that for a small number of treatments, the majority of cells come from either the ER stressed cluster of cells or the Unknown cluster of cells (*Figure 2—figure supplement 4B*). Interestingly, the treatments that led to the Unknown cluster include two HDAC inhibitors (CI 994, SAHA) (*Kraker, 2003*; *Butler et al., 2000*). It is possible that these cells are deregulated without having yet experienced a change in fate toward a more specific cell type. The treatments that lead to increased cell counts in the ER stressed cluster include two bromodomain (acetylated lysine binders) inhibitors (I-BET 151 dihydrochloride, CPI 203) (*Devaiah et al., 2012*; *Dawson et al., 2011*). While these observations do not directly help us identify better treatments for reprogramming MG, they do help us find compounds that affect the molecular state of MG cells exposed to Ascl1 overexpression (*Jorstad et al., 2017*).

## Validating sci-Plex hits in an in vivo neuronal regeneration paradigm

We next aimed to assess whether the positive hits that increased neurogenesis from the in vitro sci-Plex screen could be validated in vivo. Before testing compounds from the in vitro screen in vivo, we sought to validate that in vivo MG express the signaling pathways targeted by our small molecule compounds during neurogenic reprogramming. We analyzed scRNA-seq libraries made from our in vivo regeneration experiments where we used mice engineered to express Ascl1 specifically in the MG (*Figure 3A*), then induced retinal injury with an intravitreal injection of NMDA, and a co-injection of TSA (*Figure 3B*). The mice were sacrificed and GFP expressing cells were sorted and subjected to 10× scRNA-seq either 5, 9, or 21 days after injury the day 5 and 9 libraries were originally published in *Todd et al., 2020*; *Figure 3—figure supplement 2A, B*. Dimensionality reduction was performed to generate a UMAP from which clusters were defined and cell types were annotated using Marker genes (*Figure 3C, D*, *Figure 3—figure supplement 2A, B*). A small number of rod and cone cells are present even in the earliest time point and most likely are contaminating cells from the FACS (fluorescence-activated cell sorting). As expected, with increasing time after injury, we observe greater numbers of MG-derived bipolar cells. This is most evident in the time between 9 and 21 days. However, even

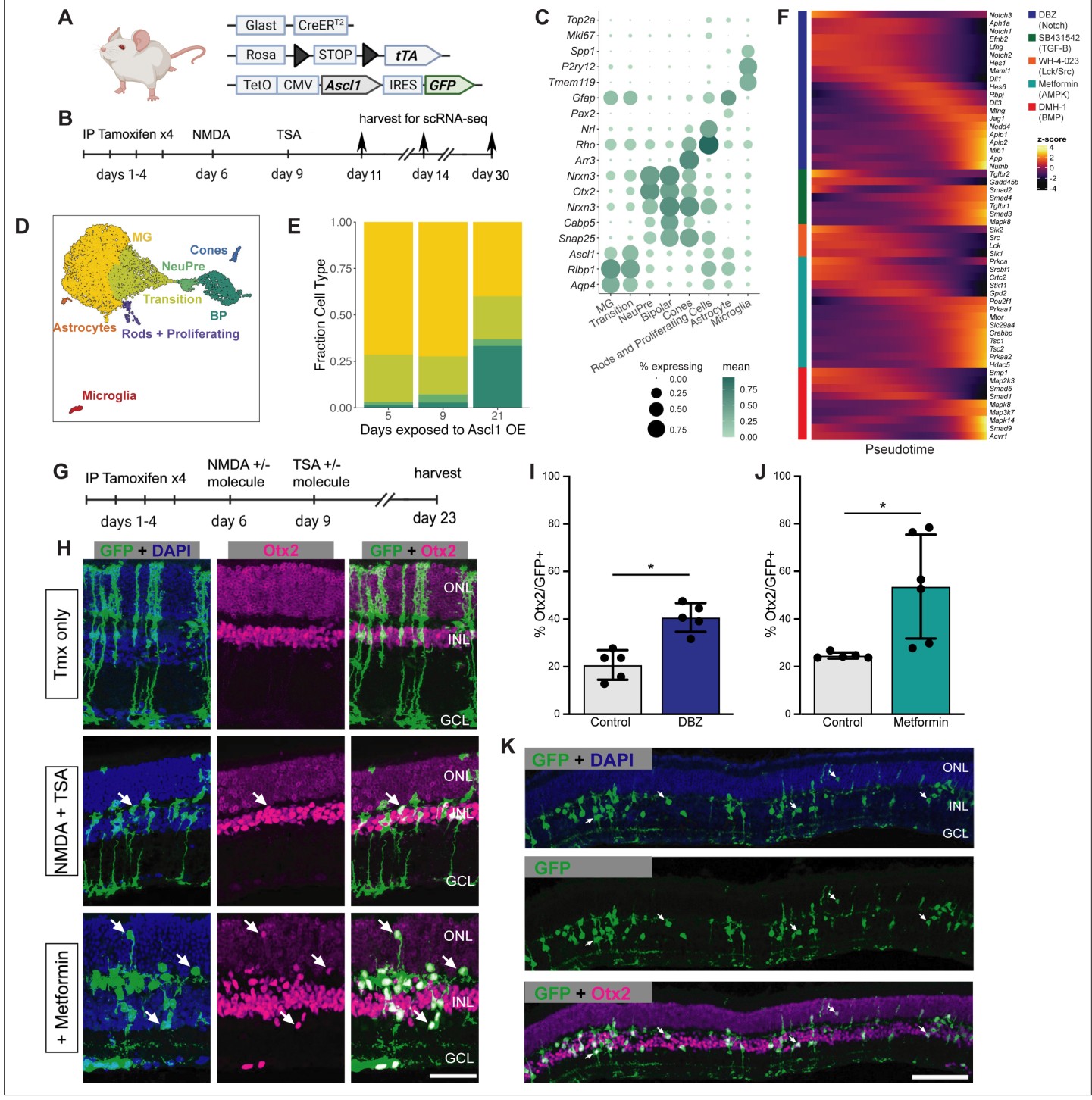

**Figure 3.** Validation of sci-Plex hits in an in vivo neuronal regeneration paradigm. (**A**) Schematic of the transgenic mouse used to induce Ascl1 and GFP specifically in Muller glia (MG). (**B**) Experimental paradigm for testing the in vivo dynamics of Ascl1-dependent MG reprogramming. (**C**) Dot plot of the genes used to define the major cell types found in the reprogramming cells in vivo across all time points. The size of the dot indicates the percent of cells that express the gene of interest. The color indicates the log10 mean UMIs per cell. (**D**) An integrated UMAP of the cells recovered from in vivo reprogramming 5, 9, or 21 days after NMDA treatment. Cells are colored by cell type. (**E**) Stacked bar plot of the cell type composition across in vivo Ascl1 OE durations. The colors represent the cell types as in D. Only the MG and MG-derived cell types are included. (**F**) Heatmap of the in vivo expression of genes related to the pathways regulated by the top small molecules hits from the in vitro screen. Reprogramming cells are ordered along pseudotime. The row normalized z-score was calculated from size factor normalized gene expression counts. (**G**) Experimental paradigm for testing the in vivo effect of hit compounds on reprogramming. (**H**) Representative sections of retina after Ascl1 only, Ascl1/NMDA/TSA or Ascl1/NMDA/TSA/

*Figure 3 continued on next page*

*Figure 3 continued*

Metformin treatment. Immunostaining for GFP (green) and Otx2 (purple) show MG-derived cells (GFP+) expressing the neuronal marker Otx2. Scale bar = 50 µm. (**I**) Quantification of the percentage of GFP+ MG-derived cells that express the neuronal marker in Ascl1/NMDA/TSA (control) versus the addition of Metformin (p = 0.017). Statistical significance (p < 0.05) as denoted by * was determined using an unpaired *t*-test. Height of the bar indicates the mean, and the error bars indicate the standard deviation. Each individual is plotted as a dot. (**J**) Quantification of the percentage of GFP+/Otx2+ cells in control versus DBZ treatment (p = 0.0009). Statistical significance (p < 0.05) as denote by * was determined using an unpaired *t*-test. Height of the bar indicates the mean, and the error bars indicate the standard deviation. Each individual is plotted as a dot. (**K**) Representative widefield image of GFP+ MG-derived neurons (GFP+/Otx2+) showing the widespread stimulation of neurogenesis in metformin treated retinas. Scale bar is 100 µm. Figure 3A was created with BioRender.com.

The online version of this article includes the following figure supplement(s) for figure 3:

**Figure supplement 1.** Quality control of in vivo reprogramming 10× RNA-seq libraries.

**Figure supplement 2.** Cell type annotation of in vivo reprogramming 10× RNA-seq libraries.

**Figure supplement 3.** Pseudotime of in vivo reprogramming 10× RNA-seq libraires.

**Figure supplement 4.** Immunostaining of in vivo reprogrammed neurons.

between 5 and 9 days there is a small increase in bipolar cells. Additionally, when pseudotime analysis was performed, a shift in the MG cluster toward a more mature state became apparent even by 9 days (*Figure 3E*, *Figure 3—figure supplement 3A, B*). Lastly, using the pseudotime analysis across all time points, we looked at how the expression of key target genes for each of the drugs are expressed throughout the reprogramming process (*Figure 3F*). In all cases, at least some of the target genes are expressed early and decrease over time, making them ideal candidates for small molecule intervention.

Using the same mice as above, we aimed to validate five hits in vivo (the γ-secretase inhibitor DBZ, Lck and Src inhibitor WH-4-023, LKB1/AMPK activator metformin, the BMP inhibitor DMH-1, and the TGF-BR1 inhibitor SB43152) for their ability to increase neurogenic reprogramming of MG in vivo. We originally also identified RepSox as a hit, but because RepSox and SB43152 have the same target, we decided to only pursue SB43152. Ascl1 was induced in the MG with tamoxifen, retinal injury was performed by NMDA, and TSA was given in conjunction with one of the small molecule hits from the screen; animals were sacrificed 2 weeks later, and their retinas were assayed for lineage traced MG-derived cells (GFP+) and the retinal neuronal maker (Otx2+) (*Figure 3G*). Out of the compounds tested, two significantly boosted the number of Otx2+ neuronal cells (*Figure 3H–J*, *Figure 3—figure supplement 4A*). The first identified compound was DBZ, a γ-secretase/Notch inhibitor (*Figure 3I*). This is consistent with reports in the chick retina where the γ-secretase inhibitor DAPT increased neuronal differentiation from proliferating MG (*Hayes et al., 2007*; *Ghai et al., 2010*; *Todd et al., 2016*). Inhibition of Notch is also well known to promote neural differentiation from retinal progenitors in mouse and human (*Finkbeiner et al., 2022*; *Nelson et al., 2007*). While this result was not unexpected, DBZ represents a new compound found to promote neurogenesis in the mammalian retina. The second positive hit, metformin, showed the biggest effect on MG-neurogenesis (*Figure 3H, I*). In some retinas, metformin increased the rate of neurogenesis to 75% of the Ascl1-expressing MG, a higher rate than any of our previous pharmaceutical treatments. The MG-derived neurons expressed Otx2 and were primarily located in the INL (inner nuclear layer); this is similar to what we observe in the ANT treatment condition, and so it does not appear that metformin altered the fates of the MG-derived neurons; In addition, the neurons in the metformin treated retinas do not express markers of ganglion/amacrine cells (*Figure 3—figure supplement 4B*). These in vivo experiments thus confirm that the in vitro sci-Plex screening is an effective method to identify new compounds that increase the regeneration potential of mammalian MG.

## Discussion

Screening large numbers of chemical compounds has become a routine path to drug discovery, but limitations remain. Most high-throughput screening assays use relatively limited read-outs, such as cell morphology, proliferation or changes in expression of a reporter. Many assays cannot access molecular information about responding cells or identify factors that make a cell type resistant to a treatment. Single-cell transcriptomic assays overcome these hurdles, but the conventional protocols are costly and highly limited in sample number. Here, we demonstrate for the first time the utility

of sci-Plex, an ultra-scalable scRNA-seq protocol, for studying reprogramming primary cells through screening drugs in low-input cultures and subsequently validating them in vivo. Our experiments both identified drugs that support reprogramming and revealed underlying biology important for the progression of glial cell reprogramming.

We first used sci-Plex to pinpoint the moment during in vitro reprogramming at which cells become committed toward a neurogenic cell fate independent of continued Ascl1 expression. Next, we tested 92 compounds across 3 doses each and identified 6 molecules of interest, two of which also had significant proneural effects in vivo. With our observation that cells no longer require Ascl1 to maintain their neuronal identity after a certain point along their reprogramming trajectory, we looked at the transcriptional regulators that were changing in their expression around this point and identified Myt1 as a potentially critical factor for commitment toward the neuronal fate. Myt1 is a zinc finger-containing DNA-binding TF that is found specifically in neuronal cells and is important in neuronal differentiation (*Vasconcelos et al., 2016*; *Bellefroid et al., 1996*; *Kim and Hudson, 1992*). In neural stem cells, the *Myt1* promoter is bound by Ascl1 and protein expression is induced following the overexpression of Ascl1 (*Vasconcelos et al., 2016*). Myt1 and its family member Myt1l function by binding to and repressing their target genes which include non-neuronal genes, progenitor genes, and regulators of Notch signaling (*Vasconcelos et al., 2016*; *Mall et al., 2017*). Myt1l has also been associated with aiding in the in vitro reprogramming of fibroblasts to neurons (*Vasconcelos et al., 2016*; *Mall et al., 2017*; *Vierbuchen et al., 2010*). However, little is known about its role in the retina, and no previous reports indicate a direct role for Myt1 in committing cells to the neuronal fate after the removal of other proneural TFs. Additional work will be needed to determine whether Myt1 is the regulator holding cells in the neuronal fate after removal of Ascl1 in our Pulse experiment, but the depth of information that can be obtained by single-cell sequencing provided us with an intriguing candidate gene.

The first hit of our screen was the Notch inhibitor DBZ. It targets the ability of gamma-secretase to cleave the intracellular domain of activated Notch receptors. While DBZ has not previously been used in retinal reprogramming studies, Notch is well known to affect neurogenesis and neural regeneration in the retina. Pharmacological manipulation of Notch signaling has been used to alter the proliferative and neurogenic capacity of MG in fish and chick retina (*Hayes et al., 2007*; *Ghai et al., 2010*; *Conner et al., 2014*; *Wan et al., 2012*). It is also interesting that Myt1, identified in our Pulse experiment as a factor that may prevent neuronal cells from returning to the progenitor fate, inhibits Notch signaling (*Vasconcelos et al., 2016*). That our screen recovered a γ-secretase inhibitor, in line with the known role of Notch in the retina, confirmed the ability of our workflow to identify important regulators.

Unexpectedly, the second compound that increased neurogenesis in vivo was metformin. Mechanistically, metformin is typically thought to work via AMPK, and while used clinically for Type2 diabetes, many studies have noted its neuroprotective and anti-neuroinflammatory effects (*Markowicz-Piasecka et al., 2017*). In our screen, metformin increased the total number of cells recovered by sci-Plex, including those in the neurogenic precursor stage. Even with the increased number of cells recovered compared to other treatments, the small number of cells recovered from each treatment by sci-Plex made it difficult to identify confident DEGs. However, by leveraging follow-up experiments in vivo we did find that metformin had a clear effect on neurogenesis from Ascl1-reprogrammed MG. The effect we find on neurogenesis is consistent with the promotion of neurogenesis reported in another in vitro study using mouse cortical and hippocampal progenitors and human ESC-derived neural progenitors (*Wang et al., 2012*), and may in part be mediated by Gadd45g, via DNA methylation (*Zhang et al., 2022*). Moreover, metformin restored blood flow and vascular density in aged mice and promoted neurogenesis from the SVZ in vivo and in vitro (*Zhu et al., 2020*). Further investigations of this molecule in astrocyte to neuronal reprogramming in other regions of the CNS will be necessary to validate the utility of metformin in in vivo neural regeneration more generally.

Repairing the nervous system has been an elusive goal, and while progress is being made with transplantation of pluripotent stem cell derived neurons, there are many complexities in neural transplantation. Reprogramming glia to serve as a source of neural repair is a strategy utilized by non-mammalian vertebrates, and inspired by nature, we and others have shown that harnessing this latent neural regenerative capacity by proneural TF over-expression can revive the neuronal reprogramming program in mammals. Layering in the use of small molecules can increase the efficiency of such reprogramming, and we show here that screening through larger libraries of compounds at single-cell

resolution in vitro can point us toward new in vivo hits. We are particularly excited by the future therapeutic potential of our compounds: metformin and DBZ; as well as the use of similar screening strategies for the identification of treatments in glial-based regeneration and wider contexts.

## Materials and methods
### Mouse injections and husbandry

Mice were housed and treated under University of Washington Institutional Animal Care and Use Committee approved (UW-IACUC: 2448-08). Animals were maintained in groups of 2–5 per cage. For the in vivo experiments, *Glast-CreER:LNL-tTA;TetO-mAscl1-ires-GFP* mice are from mixed backgrounds of C57BL/6 and B6SJF1. The tetO-mAscl1-ires-GFP mice were a gift from M. Nakafuku (U. Cincinnati), and the Glast-CreER (Jax 012586) and LNL-tTA (Jax 008603) mice were from Jackson Labs. Both sexes were used in all experiments and the mice used for injections were between the ages of P40–P60. To induce Ascl1 in MG in vivo, tamoxifen (Cayman Chemical, #13258-1G) was given intraperitoneally at 1.5 mg per 100 µl for four consecutive days. Intravitreal injections were performed on isoflurane anesthetized mice using a 32-gauge Hamilton syringe. All intravitreal injections were done at 2 µl, and contained either NMDA (Sigma, #M3262-100MG) + drug or TSA (Tocris #1406/1) + drug. NMDA injections were done in 1 µl volume at a concentration of 100 mM. TSA injections were given in 1 µl volume at a concentration of 1 µg/µl. Drugs used were all from the Tocriscreen Stem Cell Library (Tocris, 7340) and were administered in 1 µl of dimethyl sulfoxide (DMSO) at 10 mM. The mice were either randomized by animal tag number for injections (L1-4) or the vehicle (DMSO) was given to the left eye and the drug was given to the right eye (M). Blinding was performed for experiments.

| Experiment | Treatment | Plot | N |
|---|---|---|---|
| M2 | PBS (left eye) | DBZ | 3 |
| M2 | DBZ (right eye) | DBZ | 3 |
| 149 | DBZ | DBZ | 2 |
| 149 | DMSO | DBZ | 2 |
| 155 | Metformin | Metformin | 3 |
| 155 | DMSO | Metformin | 2 |
| 152 | Metformin | Metformin | 3 |
| 89 | DMSO | Metformin | 3 |
| M1 | PBS (left eye) | Non-sig | 3 |
| M1 | SB431542 (right eye) | Non-sig | 3 |
| 188 | SB431542 | Non-sig | 2 |
| 188 | DMH-1 | Non-sig | 3 |
| 168 | WH-4-023 | Non-sig | 2 |

### Primary cell culture, MG freezing, thawing, and plating

MG cultures were derived from the retinas of Postnatal Day 11 (P11) rtTa:tetO-Ascl1-ires-GFP mice of both sexes. After retina harvest, the retinas were incubated for 7 min at 37°C in a solution of papain and deoxyribonuclease (DNase) (Worthington, #LK003172). Incubation was followed by trituration of the mixture to dissociate the cells, then to stop the reaction, an equal volume of ovomucoid (Worthington, #LK003182) was added. To isolate the cells, the solution was spun at 4°C at 300 × *g* for 10 min. Pelleted cells were resuspended in growth medium consisting of Neurobasal medium (Gibco, #10888-022), 10% tet-approved fetal bovine serum (tet-FBS) (Takara/Clontech, #631367), N2 (Gibco, #17502-048), 1 mM L-glutamine (Gibco, #25030-081), 1% penicillin–streptomycin (Gibco, #15140-122), and mouse epidermal growth factor (100 ng/ml) (R&D Systems, #2028-EG-200). Cells were plated in a 6-well dish at a density of about 2 retinas per 10 cm$^2$ well, then incubated at 37°C. The entire volume of medium was changed every 48 hr until confluent or after 7 days. At confluence or after 7 days,

cells were removed from the plate using TrypLE (Gibco, #12605-028), spun at 4°C at 300 × *g* for 10 min, then resuspended in a small volume of growth medium. The entire volume of cell suspension was added to a freezing medium containing 40% tet-FBS, and 10% DMSO, with the remaining 50% being the growth medium from the resuspension. The cells were then frozen for at least 24 hr at −80°C before being moved to storage in liquid nitrogen for at least 1 day. Cells were thawed at room temperature, then spun at 300 RCF for 10 min. After thawing, cells were seeded in 500 µl of 10% tet-FBS growth medium in 24-well plates or in 250 µl of the same medium in 48-well plates. Cells were grown for 24–48 hr in the growth medium until they appeared healthy.

## Drug treatment

### Ascl1 timecourse and pulse

Once cells were confluent, the culture medium for all conditions was replaced with 1% tet-FBS-reduced growth medium. To activate the tetO-mediated transgenes, doxycycline (Sigma, #D9891-5G) was added to the media at a concentration of 2 µg/ml at various time points over the course of 5 or 7 days.

### Timecourse

Cells received doses of tet-FBS-reduced growth medium with doxycycline after 1, 2, 3, 4, or 5 days in culture. Once a treatment condition received its first dose of doxycycline, it continued to receive doses every 2 days for the remainder of the experiment. Prior to a condition's first dose of doxycycline, the cells were maintained in the reduced growth medium, which was replaced every 2 days. The control condition received no doxycycline and cultures were maintained in reduced growth medium replaced every 2 days.

### Pulse

Cells received tet-FBS-reduced growth medium with doxycycline for 1, 2, 3, 4, or 5 days. Doxycycline was added to all conditions at the same time, Day 0, and all conditions remained in culture for 5 days. The media containing doxycycline was removed at each time point over the 5 day course and replaced with 1% tet-FBS-reduced medium for the remainder of the experiment.

### Drug treated cells

Drugs were reconstituted in DMSO with a starting concentration of 10 mM and were diluted to 2.5 mM, 250 µM, and 25 µM with a mixture of 50% DMSO and 50% phosphate-buffered saline (PBS). Each well (48-well plate) of cells received a total volume of 250 µl of tet-FBS-reduced media with 1 µl of the designated treatment drug at each concentration (final concentration of drug 10, 1, and 0.1 µM). Control wells received 250 µl of tet-FBS-reduced growth medium with the appropriate, equivalent DMSO concentration. Cells were collected after 5 days with an accutase dissociation (further described in the cell collection section).

## Hashing for sci-Plex

To harvest cells for sci-Plex nuclei processing, the cells were rinsed either in 24- or 48-well plates with 200 µl HBSS (Hank's basal salt solution, Gibco, #14025-134), and then 200 µl of Accutase (Sigma, #A6964-100ML). For experiments performed in 24-well plates, the accutase was incubated on the cells an additional 1 min at room temperature before being removed. The cells were incubated with any residual accutase at 37°C for 5 min. Cells were then resuspended in 200 µl of 1% tet-FBS-reduced growth medium with a wide-bore P200 pipette tip to isolate single cells. The entire volume of the single-cell suspension was transferred to a v-bottom 96-well plate on ice. The hashing of glia used a protocol adapted from the original sci-Plex publication (*Srivatsan et al., 2020*). Briefly, dissociated cells were centrifuged at 600 × *g* for 5 min in 96-well v-bottom plates. Note that all centrifugation steps, including this one, were performed in a chilled swinging bucket centrifuge (4°C). Media was aspirated, and 200 µl 1× dPBS (no calcium, no magnesium) was added to each well. The plate was centrifuged at 600 × *g* for 5 min and the dPBS was pipetted off. To each well containing cells, 50 µl of CLB + hash solution (45 µl of Cold Lysis Buffer – 10 mM Tris/HCl pH 7.4, 10 mM NaCl, 3 mM MgCl$_2$, 0.1% IGEPAL, 1% [vol/vol, Sigma-Aldrich, I8896], SuperaseIn RNase Inhibitor [20 U/µl, Ambion,

AM2694], 1% [vol/vol] bovine serum albumin [BSA] [20 mg/ml, NEB, B9000S] + 5 µl of hash oligo [10 µM, IDT (Integrated DNA technologies)]) was added and then mixed by pipetting up and down for 5–10 strokes. The plate was incubated on ice for 3 min, after which, 200 µl of fixation buffer (5% paraformaldehyde [EMS, cat. no. 50-980-493], 1.25× dPBS (Dulbecco's Phosphate-Buffered Saline)) was mixed with the nuclei by pipetting up and down for 10 strokes. The cells were incubated with the fixative for 15 min on ice during which the hashes were affixed to the nuclei. All nuclei from all wells were then collected into a single 15 or 50 ml conical tube, depending on the size of the experiment, and centrifuged for 5 min at $800 \times g$. Pellets were resuspended in 1 ml of NSB (Nuclei Buffer + SuperaseIn + BSA – 10 mM Tris/HCl pH 7.4, 10 mM NaCl, 3 mM MgCl$_2$, 1% [vol/vol] BSA, 1% [vol/vol] SuperaseIn RNase Inhibitor), and centrifuged again for 5 min at $800 \times g$. One final wash with 1 ml of cold NSB was performed. Upon resuspension, nuclei were transferred to a 1.5-ml LoBind microcentrifuge tube (Eppendorf, Z666491) and counted using a hemocytometer. The 1.5 ml tube was placed within a 15-ml conical and centrifuged in a swinging bucket centrifuge for 5 min at $800 \times g$. The nuclei were resuspended in 500 µl of NSB and flash frozen in liquid nitrogen before being stored at −80°C.

## Sectioning and immunostaining

For immunohistochemistry, whole eye globes were fixed for 30 min in 4% paraformaldehyde after lens removal. Eyes were then soaked in 30% sucrose overnight and processed for sectioning the following day. For sectioning, retinas were frozen in O.C.T. and cryosectioned at 18 µm thickness. Slides were washed in PBS two times for 10 min and then primary antibodies were applied overnight in 10% normal horse serum (Vector Labs, #S-2000-20), 0.5% Triton X-10 (Sigma, #T8787-100ML), and PBS. The following day sections were rinsed two times for 10 min in PBS and secondary antibodies were applied at a concentration of 1:1000 in PBS for 1 hr. Slides were washed again in PBS and then coverslipped using Fluromount-G (Southern Biotechnology, #0100-01).

| Antibody/stain | Source | Catalog # | RRID | Dilution |
|---|---|---|---|---|
| Primary | | | | |
| Chicken anti-GFP | Abcam | Cat#: Ab13970 | RRID: AB_300798 | 1:1000 |
| Goat anti-OTX2 | R&D Systems | Cat#: BAF1979 | RRID: AB_2157171 | 1:500 |
| Mouse anti-HuC/D | Invitrogen | Cat#: A-21271 | RRID: AB_221448 | 1:500 |
| Secondary | | | | |
| Donkey anti-chicken 488 | Jackson Immuno | Cat#: 703-545-155 | RRID:AB_2340375 | 1:1000 |
| Donkey anti-mouse 568 | Life Technologies | Cat#: A10037 | RRID:AB_2534013 | 1:1000 |
| Donkey anti-goat 568 | Life Technologies | Cat#: A11057 | RRID:AB_2534104 | 1:1000 |
| Stain | | | | |
| DAPI | Sigma | Cat#: D9542 | NA | 1:7700 |

## Microscopy and cell counts

A Zeiss LSM880 confocal microscope was used to take images from the in vivo experiments. For quantifications, four images per retina at ×20 objective were taken and both eyes were combined for a biological n. Statistical significance between treatments was determined using an unpaired $t$-test.

## 10× library preparation

Dissociated and GFP+ FACs sorted cells prepared 21 days after NMDA injury were subjected to the standard 10× workflow for Single Cell 3′ v3.1 Dual Index Gene Expression kit (Dual Index Kit TT Set A 96 rxns, 10× Genomics, 1000215).

## sci-RNA-seq

After hashing, the nuclei were subjected to the original 2-level sci-RNA-seq protocol (*Cao et al., 2017*) with minor modifications. Briefly, the hashed nuclei were thawed on ice, subjected to centrifugation at $800 \times g$ for 5 min, the liquid was aspirated away, and then the nuclei were incubated on ice

for 3 min in 500 µl of NSB + 0.2% Triton X-100 (Thermo Fisher, A16046.AP). Permeabilized nuclei were centrifuged for 5 min at 800 × g, resuspended in 400 µl NSB, and counted using a hemocytometer. The desired number of nuclei was subjected to one final centrifugation step at 800 × g for 5 min. They were then resuspended to a concentration of ~800–3000 nuclei per µl across 3–4× twin.tec 96 Well LoBind PCR Plates (Eppendorf, 0030129512) in NSB. The RT reaction was carried out using an increasing temperature gradient (4°C 2 min, 10°C 2 min, 20°C 2 min, 30°C 2 min, 40°C 2 min, 50°C 2 min, 55°C 15 min), but was otherwise the same as in Cao et al.

All nuclei were pooled. DAPI (4',6-diamidino-2-phenylindole, 3 µM final) (Invitrogen, D1306) was used to stain the DNA content of cells so that doublets and debris could be removed by sorting on the DAPI height versus DAPI area, and FSC (forward scatter) versus SSC (side scatter), respectively, using a FACSAria III cell sorter (BD Biosciences). Into each well of a twin.tec 96 Well LoBind PCR Plate, ~180 nuclei were sorted. This was performed across four to five plates depending on the experiment. Each of the wells that the nuclei were sorted into contained 5 µL of EB buffer (QIAGEN, 19086), 0.5 µl of 5× mRNA Second Strand Synthesis buffer (New England Biolabs, E6111L), and 0.25 µl of mRNA Second Strand Synthesis enzyme (New England Biolabs, E6111L). The plates were incubated at 16°C for 3 hr. Tagmentation was performed at 55°C for 5 min in 5.75 µl of Tagmentation mix 1.1 µl N7 loaded custom TDE1 enzyme (MacroLab, UC Berkeley) per 632.5 µl 2× TD (DNA tagmentation) buffer (20 mM Tris–HCl pH 7.6, 10 mM MgCl$_2$, 20% vol/vol dimethyl formamide [Sigma-Aldrich, 227056-100ML]). The reaction was stopped with 12 µl of DNA binding buffer (Zymo Research, D4004-1-L). Ampure-based bead purification, PCR, and final library purification were performed as published previously (*Cao et al., 2017*). The library was visualized using a D1000 Screen Tape (Agilent Technologies, 5067-5583) and quantified by Qubit using Broad Range DNA reagents (Invitrogen, Q32853).

In-depth protocols can be found on protocols.io under the title.

Single-cell RNA sequencing library preparation (2-level sci-RNA-seq).

## Sequencing analysis

### sci-Plex initial processing

The sci-Plex libraries were sequenced on an Illumina Nextseq550 (High Output 75 cycle kit) with 18 cycles for Read1, 10 cycles for each index, and 52 cycles for Read2. The reads were demultiplexed and then aggregated into individual cells using a pipeline developed by the Brotman Baty Institute (BBI) which is available at the bbi-lab github page under the bbi-dmux and bbi-sci repositories (https://github.com/bbi-lab/bbi-dmux [*Pliner, 2024a*]; https://github.com/bbi-lab/bbi-sci [*Pliner, 2024b*]). Custom code was used to incorporate the hash information outputted by the pipeline into a Monocle3 object.

### Filtering and dimensionality reduction (Pulse vs Timecourse)

Cells with <500 UMIs and >20,000 UMIs were discarded. The ratio of the most common hash barcode to the second most common hash barcode was calculated (top_to_second_best_ratio). For the time course experiment, all cells with a top_to_second_best_ratio > 4 were retained. A cutoff of a top_to_second_best_ratio > 5 was used for the pulse experiment. Using Monocle3, the two datasets were combined with `combine_cds()` and subjected to dimensionality reduction to generate a 2D UMAP from 20 PCA dimensions, a UMAP mindist of 0.1, and 50 nearest neighbors. To best align the two datasets, `align_cds(alignment_group = "experiment", residual_model_formula_str = "~log10(n.umi)", preprocess_method = "PCA")` was applied after `preprocess_cds()` but before `reduce_dimension()` in the standard Monocle3 pipeline. Clustering was performed with a resolution of $6 \times 10^{-4}$, and cell types were assigned using marker genes (*Supplementary file 1a*).

### Pseudotime analysis (Pulse vs Timecourse)

Pseudotime scores were calculated with the root node set to the node at the center of the MG cluster. DEG analysis (`fit_models()`) was performed across pseudotime for cells with a pseudotime score of >10 and <20. Genes with an effect size of magnitude <0.08 were filtered out. The remaining 250 genes with the lowest *q*-value were used in downstream analyses (Dataset S1). Expression heatmaps were constructed with ComplexHeatmap (*Gu et al., 2016*; *Gu, 2022*) using all cells with a non-infinite

pseudotime score. Cells were ordered by their pseudotime score and z-scores of the `normalized_counts()` are displayed.

### Filtering and dimensionality reduction (small molecule screen)

For both Collections 1 and 2, cells with <500 UMIs, >20,000 UMIs, or a top_to_second_best_ratio < 5 were discarded. Additionally, any cells with >15% mitochondrial reads were removed from downstream analyses. The data from Collections 1 and 2 were combined as above and then subjected to dimensionality reduction with 30 PCA dimensions, a mindist of 0.1, and 100 nearest neighbors. Clustering was performed with `cluster_cells()` using a resolution of $1 \times 10^{-4}$. Marker genes were used to annotate cell types (**Supplementary file 1a**).

### Sub clustering of reprograming MG cells (small molecule screen)

The cells annotated as Bipolar and Reprogramming_MG were selected and dimensionality reduction was performed of the subset of cells with 30 PCA dimensions, a mindist of 0.1, and 12 nearest neighbors. A resolution of $5 \times 10^{-4}$ was used for clustering. Common markers were used to assign cell types on a cluster by cluster basis (**Supplementary file 1a**), but some clusters could not be assigned in this manner. To find the sets of genes that most specifically define these unknown clusters, for each gene, the fraction of cells within a cluster that express the gene as well as the mean expression of that gene within a cluster were calculated. The mean expression values were then normalized to sum to one across all cell types for a given gene. This was used as input for Jensen Shannon distance calculations comparing the observed normalized expression distribution across cell types to simulated distributions in which all expression of a given gene comes from a single-cell type. This was done for each cell type across all genes. Specificity scores were calculated by subtracting the Jensen Shannon distance from 1. The genes with a fraction expressing >0.05 and a specificity of >0.3 were categorized as highly specific for that cell type. For each Unknown cluster, the highly specific genes were used as input to `enrichGO()` from the clusterProfiler package with a pvalueCutoff of 0.01, qvalueCutoff = 0.05, and Benjamini–Hoechberg adjustment using the org.Mm.eg.db OrgDb. The enriched biological processes (BP) were then displayed by barplot.

### Cell type quantification and fold-change analysis

Heatmaps of cell type abundance across treatments were generated with `complexHeatmap()`. For fold-change calculations, the BP and NeuPre cell types were merged (Neuronal), as were the ProL (early) and ProL (late) cells (ProL). Any treatments with fewer than 20 cells recovered were discarded. A cell type by treatment matrix was generated and `size_factor()` adjusted. The matrix was used to calculate the fold change in cell counts of each treatment compared to the Ascl1 only condition for all cell types. The total cell counts from each condition were also tabulated.

### 10× initial processing

The 10× libraries were sequenced on an Illumina Nextseq500 (150 cycle kit) with 28 cycles for Read1, 10 cycles for each index, and 90 cycles for Read2. The reads were processed using CellRanger (3.1.0) (**Zheng et al., 2017**) with the default settings. The output was imported into Monocle3 (1.0.0) (**Cao et al., 2019**), and the D21 data was combined into a single object with the previously published D5 and D9 datasets (**Todd et al., 2020**).

### Filtering and dimensionality reduction (in vivo)

Cells with <2500 UMIs (D5 and D21) or <3000 UMIs (D9) were discarded. Using Monocle3, the three datasets were combined with `combine_cds()` and subjected to dimensionality reduction to generate a 2D UMAP from 20 PCA dimensions, a UMAP mindist of 0.1, and 50 nearest neighbors. To best align the two datasets, `align_cds(alignment_group = "sample", residual_model_formula_str = "~log10(n.umi)", preprocess_method = "PCA")` was applied after preprocess_cds() but before `reduce_dimension()` in the standard Monocle3 pipeline. Clustering was performed with a resolution of $6 \times 10^{-4}$, and cell types were assigned using marker genes (**Supplementary file 1a**).

## Pseudotime analysis (in vivo)

Pseudotime scores were calculated with the root node set to the node at the center of the MG cluster. Expression heatmaps were constructed with ComplexHeatmap (*Gu et al., 2016*; *Gu, 2022*) using all cells with a non-infinite pseudotime score. Cells were ordered by their pseudotime score and z-scores of the `normalized_counts()` are displayed.

## Resource availability

### Lead contact

Further information and requests for resources and reagents should be directed to and will be fulfilled by the lead contact, Thomas A. Reh (tomreh@uw.edu).

### Materials availability

No unique materials were generated by this study.

## Acknowledgements

We would like to thank the members of the Trapnell and Reh lab for their valuable comments and discussions. We thank Choli Lee for assistance in flow sorting, and the Brotman Baty Institute Advanced Technology Lab for support with the data processing pipeline. Funding: this work was supported by the National Institutes of Health (1R01HG010632 to CT; R01EY021482-12 to TAR; K99EY033402 to LT and 1F32EY032331 to AT), the Paul G Allen Frontiers Foundation (Allen Discovery Center grant 12357 to CT), the Chan Zuckerberg Initiative (CZF2019-002442 to CT), the Foundation Fighting Blindness (TA-RM-0620-0788-UWA to TAR), and the International Retina Research Foundation Fellowship to MH. Illustrations were created with BioRender.com.

## Additional information

### Competing interests

Cole Trapnell: SAB member, consultant and/or co-founder of Algen Biotechnologies, Altius Therapeutics, and Scale Biosciences. Thomas A Reh: SAB member, consultant and/or co-founder of Tenpoint Therapeutics. The other authors declare that no competing interests exist.

### Funding

| Funder | Grant reference number | Author |
| --- | --- | --- |
| National Eye Institute | R01EY021482-12 | Thomas A Reh |
| National Eye Institute | K99EY033402 | Levi Todd |
| National Eye Institute | 1F32EY032331 | Amy Tresenrider |
| Foundation Fighting Blindness | TA-RM-0620-0788-UWA | Thomas A Reh |
| Chan Zuckerberg Initiative | CZF2019-002442 | Cole Trapnell |
| International Retina Research Foundation | Fellowship | Marcus Hooper |
| National Human Genome Research Institute | 1R01HG010632 | Cole Trapnell |
| Paul G. Allen Family Foundation | Allen Discovery Center grant 12357 | Cole Trapnell |

The funders had no role in study design, data collection, and interpretation, or the decision to submit the work for publication.

## Author contributions
Amy Tresenrider, Levi Todd, Conceptualization, Formal analysis, Investigation, Visualization, Methodology, Writing – original draft, Writing – review and editing; Marcus Hooper, Conceptualization, Formal analysis, Investigation, Methodology, Writing – review and editing; Faith Kierney, Nicolai A Blasdel, Investigation; Cole Trapnell, Thomas A Reh, Supervision, Funding acquisition, Writing – review and editing

## Author ORCIDs
Amy Tresenrider (ID) http://orcid.org/0000-0002-0819-9187
Levi Todd (ID) http://orcid.org/0000-0003-2561-7675
Thomas A Reh (ID) https://orcid.org/0000-0002-3524-0886

## Ethics
Mice were housed and treated under University of Washington Institutional Animal Care and Use Committee approved (UW-IACUC).

Reviewer #1 (Public Review): https://doi.org/10.7554/eLife.92091.3.sa1
Reviewer #2 (Public Review): https://doi.org/10.7554/eLife.92091.3.sa2
Author response https://doi.org/10.7554/eLife.92091.3.sa3

# Additional files

## Supplementary files
• MDAR checklist
• Supplementary file 1. Marker genes used for cell type annotation.

## Data availability
The single-cell sequencing data have been deposited at GEO: GSE239731. All code used for the analysis has been uploaded to github at https://github.com/atresen/sci-plex_glia/tree/main (copy archived at *Tresenrider, 2024*).

The following dataset was generated:

| Author(s) | Year | Dataset title | Dataset URL | Database and Identifier |
| --- | --- | --- | --- | --- |
| Trapnell C, Reh TA, Tresenrider A, Hooper M, Todd L, Kierney F, Blasdel N | 2024 | A multiplexed, single-cell sequencing screen identifies compounds that increase neurogenic reprogramming of mammalian Muller glia | https://www.ncbi.nlm.nih.gov/geo/query/acc.cgi?acc=GSE239731 | NCBI Gene Expression Omnibus, GSE239731 |

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
