## [Editor Report · eLife assessment]

This manuscript used the sci-Plex system for screening compounds to improve the Ascl1-induced reprogramming from Müller glia to bipolar neurons in vitro, followed by in vivo characterization of two promising compounds in mice. The findings are **valuable** for future studies to develop cell replacement strategies for treatment of retinal degeneration. The strength of evidence is **solid**, featuring a scalable drug screening design, albeit with limited mechanistic insights.

---

## [Referee Report · Reviewer #1 (Public Review)]

Summary:

The study used the sci-Plex system to perform in vitro screen of chemicals and found that 2 compounds improved the reprogramming efficiency in Ascl1-overexpressed MG (Muller glia), and in addition, administration of the identified compounds in the previously established in vivo model (Ascl1, NMDA, TSA) showed that DBZ and metformin increased Otx2+ cells for improved neurogenesis.

Strengths:

The overall study was straightforward and well-designed. The method in the study could be potentially useful for large-scale in vitro screens for compounds to further improve reprogramming efficiency. The data and results of the study are of good quality.

Weaknesses:

Future studies may help provide more in-depth mechanistic examinations of the reprogramming process such as whether the compound treatment indeed affects the corresponding signaling pathways.

---

## [Referee Report · Reviewer #2 (Public Review)]

Summary:

In the current manuscript, Tresenrider et al., present their recent study focusing on screening of small molecules to enhance the conversion from Müller cells (MG) to retina neurons induced by ectopic Ascl1 expression.

Strengths:

To analyze results from multiple treatment conditions in a single experiment, the authors employed a method called sci-Plex to perform scRNA-seq on mixed samples to investigate the effects of different durations of Ascl1 expression and screen for potential small molecules to promote reprogramming. Ultimately, they identified two compounds with intended activities on mouse retina. The findings may aid in future development of a cell replacement strategy for treating retinal degeneration.

Weaknesses:

The mechanistic insights are limited. Certain claims are confusing or superficial at this point.

---

## [Author Response]

The following is the authors’ response to the original reviews.

We greatly appreciate the recommendations of the reviewers and have performed further analyses with existing data where requested.

Below are our responses to each of the individual points.

**Reviewer #1 (Recommendations For The Authors):**
(1) P11 mouse retina is still quite young, would MG isolated from adult retina be more interesting and relevant to disease-oriented cell replacement therapy? How efficiently would the sci-Plex system work for in vitro screen of mature murine MG?

Thank you for bringing this up. While a protocol for the conversion of MG to neurons with adult mice in vivo exists, it has proven to be more difficult to maintain adult MG in dissociated cell cultures, due to their more limited proliferation in vitro. This makes it difficult to use the sci-Plex assay, since cell number is limiting for treatment conditions. Therefore, we have chosen the strategy of screening on P11, where MG undergo proliferative cell divisions in dissociated cultures, allowing us to grow the millions of cells needed for this assay, and then to test the efficacy of the compounds we find from the screen with an adult in vivo assay.

(2) The study identified and tested the compounds individually, how would a combination of the compounds work in vivo? It would be interesting to examine how different combinations may affect the reprogramming efficiency and neuronal compositions.

We agree that this would be very interesting to investigate. However, the number of treatment conditions then expands beyond the scale of the current sci-Plex technology with the number of MG that we are able to collect. We instead adopted the strategy of casting a very wide net to identify additional molecular pathways that might be important in the reprogramming process.

(3) In-depth mechanistic and/or functional studies of the reprogrammed MG are highly desirable to improve the quality and significance of the study and to better understand how the compounds may influence the signaling and the reprogramming process.

While we agree that this would strengthen the study, this would increase the scope of the required revisions considerably. We are very interested in following up on some of the hits and look forward to providing additional details of mechanisms in future publications. However, we feel that reporting this method and the results will stimulate those interested in reprogramming glia in other areas of the nervous system to test the compounds we identified in this assay.

**Reviewer #2 (Recommendations For The Authors):**
(1) The authors employed two protocols to initiate direct reprogramming of MG into retinal neurons in vitro. These protocols, referred to as "Timecourse" and "Pulse," involved short-term treatments lasting no more than 5 days. However, the findings obtained indicate that these brief treatments were insufficient to achieve a stable conversion. This conclusion is supported by the comparison between the "4 days (Timecourse)" and "4 days (Pulse)" conditions, as depicted in Figure 1 (D and E). In this set of experiments, labeling cells that express specific neuronal markers as neurons raises concerns, as these cells may have multiple fates, either died, reverted, arrested in certain intermediate stages, or converted to functional neurons. It is thus critical to determine whether the conversion to functional neurons is enhanced.

We thank you for your concern about this. We aimed to be very careful in our naming. In our naming scheme for this figure, we only consider the small number of cells with specific Bipolar markers (Trpm1, Grm6, Capb5, Otx2) neurons based on previous publications ((Jorstad et al. 2017; Todd et al. 2021; Todd et al. 2022; Todd et al. 2020)). The other cells that have some neuronal markers are identified as neuronal precursors (NeuPre) and are, as you mentioned, not necessarily mature/functional. While these NeuPre cells may eventually have multiple fates/may die/may revert to more ProL cells at some rate we believe it’s fair to define them as Neuronal Precursors due to the genes they are expressing (Dcx, Snap25, Elavl3, Gap43) at the moment of collection.

Furthermore, your statement indicating that “the findings obtained indicate that these brief treatments were insufficient to achieve a stable conversion” is not what we intended to demonstrate. The text will be reworked to reflect what we hoped to convey. We acknowledge that (1) the majority cells are not stably converted, and (2) the levels of NeuPre cells are lower in the Pulse experiment overall, but this is true even at Day 5 when the conditions should be the same across experiments. The Pulse and Timecourse experiments were done on different days, and having previously found that there are differences in MG to BP conversion rate from experiment to experiment, these results were not unexpected. Of more note to us was that while ProL cells, Transition cells, and MG have very different patterns of abundance across time when comparing the experiments, the NeuPre cells accumulate at a similar time and pattern across the two experiments. This indicated to us that they uniquely have some amount of Ascl1 independent stability in their cell fate even when exposed to Ascl1 for as little as 3 days. See Author response image 1 below. This plot will be added to Fig. S1.

(2) The authors made a claim that a pseudo time value of 15 represents a crucial timepoint where the transition in cell fate becomes stable and ceases to rely on ectopic Ascl1 expression. However, it is essential to provide concrete evidence to substantiate this assertion. It is prudent to perform quantitative analyses rather than relying solely on the deduced trajectory to make this claim.

This is a fair point, the value of 15 was estimated by eye. We have returned to the data and estimated a density function for the pseudotime scores of the cells from the 1, 2, 3, and 4 day conditions in both the Pulse and Timecourse experiments (Author response image 2A-B below). We then calculated 16 to be the local minima between the pseudotime values of 10-20 for the Pulse experiment (Blue line). When comparing the two experiments, it’s apparent that there is a massive accumulation of cells with a pseudotime value just lower than 16 in the Timecouse experiment (values 10-15), and very few cells across the same region for the Pulse experiment, indicating some dependence on continued Ascl1 expression for the cell fate that exists from pseudotime 10-16 (mostly ProL cells). To the contrary, cells with greater pseudotime values exist across both experiments at similar levels.

We have also looked at the expression of Ascl1 along the pseudotime trajectory in the Timecourse experiment. Interestingly, and consistent with experiments in previous studies, both in vitro and in vivo (Todd et al. 2021; Todd et al. 2022; Todd et al. 2020), we see a decrease in Ascl1 expression as the cells move towards the end of the pseudotime trajectory (C below). It’s intriguing to us that the downregulation also happens right after a pseudotime value of 16. The temporal coalescence of the loss of Ascl1 expression in the Timecourse experiment with the persistence of cells with pseudotime values > 16 in the Pulse experiment provides strong evidence that we have identified the point at which cells stop expressing Ascl1 while maintaining more mature cell fates. The plots below will be added to the manuscript.

**Author response image 2. sa3fig2:** 

(3) It is intriguing to observe that the expression of Ascl1 was down-regulated in both neuronal precursors and bipolar cells in the mouse retina following tamoxifen and NMDA treatment (refer to Fig. 3C). However, the expression of ectopical Ascl1 should have been constitutively activated by tamoxifen. Therefore, if the GFP+ bipolar cells and neuronal precursors were indeed converted from Müller cells, we would expect to capture a high level of Ascl1 expression. How to account for this discrepancy? How is the expression exogenous Ascl1 expressed from a constitutive promoter attenuated?

As discussed above, this has been observed previously. Ascl1 driven from the TTA transgenic mouse line is high in the MG, but declines as these cells are reprogrammed into neurons in vivo or in vitro. One possibility is that the TTA is not as active in neurons as in MG, but in other lines of transgenic mice, eg. TRE-Atoh1 mice, the transgene continues to be expressed at a high level even in the differentiating neurons, so this downregulation appears to be unique to Ascl1. We do not understand why Ascl1 levels decline in the differentiating neurons, but this has been a consistent finding across several studies of in vivo and in vitro reprogramming.

(4) Exogenous Ascl1 was shut down after other neuronal specific genes were induced during MG reprogramming in vitro. Is this also the case during Ascl1-mediated reprogramming in vivo? If so, do converting cells show a distinct gene expression program if exogenous Ascl1 is constitutively overexpressed?

Yes, as can be seen in Fig 3C Ascl1 expression is high in the MG and Transition cell populations, but decreases in the NeuPre and Bipolar cells. As stated above, continued high Ascl1 expression keeps cells in a more progenitor-like state. This is true in vivo and in vitro. It has been more clearly addressed upon revision.

(5) As previously documented in their Science Advances publication, the authors have established the requirement of NMDA injury for facilitating the successful induction of neuronal conversion through Ascl1 over-expression. Why is injury required for MG conversion in vivo, but not in vitro? This is related to question #1 above that certain signals may be required for the full conversion process, not just the initial induction of a few neuronal specific genes.

While the in vitro and in vivo systems share similarities, there are key differences, which affect what must be done to the cells in order to produce converted neurons. In our initial publication demonstrating that Ascl1 can reprogram mouse MG to a neurogenic state, we carried out our experiments in dissociated cell cultures (Pollak et al 2013) like those described in this report. At that time, we did not need to add either NMDA or TSA to the cultures to induce neurogenesis from Ascl1. However, when we attempted the reprogramming in vivo, we found that after postnatal day 8, injury and TSA were required in vivo (Ueki et al; Jorstad et al). We surmise that the massive neuronal loss that occurs in establishing dissociated MG cultures replaces the NMDA injury we carry out in vivo.

To your second point about the requirement for more than “just the initial induction of a few neuronal specific genes”. This is definitely true. When we carry out reprogramming in vivo with Ascl1 or other transcription factors, the MG-derived neurons acquire neuronal morphology, develop neuron-like electrophysiological properties, integrate into the retinal circuit and respond to light stimulus; however, they are still not identical in gene expression or morphology to normal retinal neurons. This is why we are continuously looking for more compounds or conditions that can help improve the process.

(6) The discovery that Metformin acts as a stimulator for MG-to-neuron conversion is interesting.However, before drawing definitive conclusions, several questions need to be addressed:(a) As specific small molecules have been identified to change cell fates, the question is whether Metformin and other effective compounds can function alone or have to effect in conjunction with Ascl1? This can and should be tested in vitro by simply treating MG with Metformin but not doxycycline.

To our knowledge there are no convincing in vivo trials in which neurons have been generated from MG using only combinations of small molecules. Because Metformin was identified in vitro due to the increase in recovered cells and not an increase in % neurons, we especially doubt it would have the desired increase in neurons without expression of a transcription factor.

(b) Metformin is known to target AMPK, but this is unlikely the only target of the drug. Does AMPK knockdown have the same enhancement effect?

In the drug screen, we also tested the AMPK inhibitor Dorsomorphin dihydrochloride, but it didn’t have any effect. However, Metformin is an activator, so it would be interesting to see in future studies if Dorsomorphin dihydrochloride could inhibit the effect of Metformin or if the enhancement is acting independently.

(c) Is the effect of Metformin specific for Ascl1 or any TF(s) that stimulates MG-to-neuron conversion?

We would like to follow up with this in future.